# Telehealth use among pediatric Alabama Medicaid enrollees, March-December 2020: Variations by race/ethnicity & place of residence

**Bisakha Sen**[1‡]*, **Md Jillur Rahim**[1‡], **Julie McDougal**[1], **Pradeep Sharma**[1], **Nianlan Yang**[2], **Anne Brisendine**[1], **Ye Liu**[1], **Van Nghiem**[1], **David Becker**[1]

1 Department of Health Policy & Organization, University of Alabama at Birmingham, Birmingham, Alabama, United States of America, 2 Clinical Research Group, PPD Inc., Austin, Texas, United States of America

‡ BS and MJR are Joint first authors on this work.
* bsen@uab.edu

## Abstract

During the early days and months of the COVID-19 pandemic, healthcare facilities experienced a slump in non-COVID-related visits, and there was an increasing interest in telehealth to deliver healthcare services for adult and pediatric patients. The study investigated telehealth use variation by race/ethnicity and place of residence for the pediatric enrollees of the Alabama Medicaid program. This retrospective observational study examined Alabama Medicaid claims data from March to December 2020 for enrollees less than 19 years. There were 637,792 pediatric enrollees in the Alabama Medicaid program during the study period, and 16.9% of them had used telehealth to meet healthcare needs. This study employed a multivariate Poisson mixed-effects model with robust error variance to obtain differences in telehealth utilization and found that Non-Hispanic Black children were 80% as likely, Hispanic children were 55% as likely, and Asian Children were 46% as likely to have used telehealth compared to Non-Hispanic White children. Pediatric enrollees in large rural areas and isolated areas were significantly less likely (IRR: 0.90 for both, p<0.05) to use telehealth than those in urban areas. This study's findings suggest that attention needs to be paid to addressing race/ethnicity disparities in accessing telehealth services.

## Introduction

It is well-known that telehealth became a crucial vehicle for healthcare delivery to children and adults alike during the COVID-19 pandemic. In the U.S., under the Public Health Emergency (PHE) declaration, the Centers for Medicare and Medicaid Services (CMS) extended coverage eligibility for telehealth services and eased regulatory requirements for telehealth, and states and private health insurers followed suit. The PHE has currently been extended through November 2022 [1]. However, the U.S. House of Representatives has already voted with bipartisan support to extend Medicare telehealth benefits through 2024 [2], and as of July 2022, 21

**Data Availability Statement:** This research was conducted using administrative claims data from Alabama Medicaid that includes PHI, which we

accessed through a research contract with Alabama Medicaid. We are prohibited from publicly making this data available. However, interested researchers can contact us for the data after obtaining written permission from the Alabama Medicaid Agency to access the data (https://medicaid.alabama.gov/).

**Funding:** This study was funded in part by the Alabama Medicaid Agency (https://medicaid.alabama.gov/), contract number: C200629944. BS is the principal investigator of the award. The funders had no role in study design, data collection and analysis, decision to publish, or preparation of the manuscript.

**Competing interests:** The authors have declared that no competing interests exist.

states had implemented policies at the state-level requiring payment parity between services delivered via telehealth and in-person, and five more states had implemented payment parity with caveats [3]. Hence, telehealth is on track to remain a major vehicle for healthcare delivery post-PHE. Thus, questions about disparities in telehealth access and use, and the extent to which it benefits vulnerable populations, are of paramount interest.

Findings on disparities in telehealth use among adults are mixed. Surveys of adult respondents indicate that 17%-23% of adults have used telehealth for healthcare needs [4, 5], with higher use reported among respondents of color and lower-income households. At the same time, reports from several healthcare systems suggest that patients of color, rural patients, and publicly insured patients are less likely to use telehealth. [6–8] Prior studies also find that the association of patient characteristic like education with accessing routine healthcare vary by race/ethnicity [9]. For pediatric patients, there is evidence of telehealth visits increasing across a range of outpatient specialties and cautious optimism that telehealth will help disadvantaged pediatric populations by reducing "parent burden, decreasing both times off from work and distances traveled for health care" [10]. However, there is a paucity of population-level research on pediatric telehealth use, which informs on inequities by race/ethnicity and place of residence. This study helps address that gap by examining telehealth use among the pediatric enrollee population in Alabama's Medicaid Program. Like other Deep South states, Alabama is characterized by high poverty, large African-American populations, a substantial rural population, and poor performance on health indicators [11–13]. Thus, this study can inform on telehealth use among a particularly vulnerable sub-population—low-income children in one of the most disadvantaged regions of the U.S.

## Materials and methods

This retrospective observational study used claims data from Alabama Medicaid for enrollees aged 0–18 years who were enrolled for any period of time from March to December 2020. Telehealth claims were identified using the place of service code of 02 on the medical claims data.

Enrollee characteristics like race/ethnicity, age, self-reported gender, and county and zip code of residence were extracted from Medicaid enrollment files. Telehealth use from March to December 2020 was examined for the demographic characteristics of enrollees. Race/ethnicity was categorized into non-Hispanic white (NHW), non-Hispanic Black (NHB), Hispanic, Native American, Asian, and Other (which includes unknown). [14, 15] Rural-Urban Commuting Area (RUCA) designation, a standardized classification system to identify rural and urban areas, was divided into four categories: urban, large rural, small rural, and isolated areas. [16] The latest version of RUCA codes, revised on July 2019, was obtained from the United States Department of Agriculture. [17] To measure the socio-economic status of the area where the enrollee resided, quartiles of zip-code level poverty were constructed (Pov_Q1 through Pov_Q4), following precedence in the literature. [18, 19] To construct this, the percentage of households living below the federal poverty line at each zip code level was collected from the US census bureau, and divided into quartiles [20]. Note that lower quartiles indicate a smaller share of households in poverty. A binary indicator indicating whether the zip code was in the highest decile nationwide for lack of broadband access versus all other deciles. In the highest decile, a minimum of 40.2% and an average of 53.7% of zip-code residents reported no broadband or data access, compared to an average of 19.5% reporting no broadband access in the other deciles pooled.

The outcome of interest was a binary indicator for whether an enrollee used any telehealth services from March to December 2020. Descriptive statistics were computed on the rate of telehealth use by the above enrollee characteristics. Given the possibility of correlation between

RUCA, zip-code level poverty, and lack of broadband access, multicollinearity between variables was tested by using a linear probability model and computing and estimating variance inflation factors.

Since one goal of this study is to investigate the association between neighborhood characteristics and telehealth utilization, a Poisson mixed effects model with robust error variance was estimated with zip code level as the nested structure. The Poisson mixed effects model incorporates random effect for each level of hierarchy to account for correlation among observations at each level and accounts for overdispersion, a situation when variance is greater than the mean, that could lead to a biased estimate. [21, 22] Due to the relatively large sample size of Medicaid enrollees, even minuscule differences by enrollee characteristic may appear to be statistically significant; hence meaningful effect size of differences in telehealth use was of interest rather than just whether the differences were statistically significant. To this end, given the well-known challenges of obtaining effect sizes from odds ratios from logistic regressions and the documented distortions of scientific findings when odds ratios have been misinterpreted as risk ratios [23, 24], the Poisson models were preferred, since the incidence rate ratios from such models can be interpreted as relative risk ratios [25]. Additional controls for potential confounders included in the multivariate Poisson mixed effects model were self-reported gender, age group (0–3 years, 3–6 years, 6–12 years, and 12–18 years), pediatricians or family medicine practitioners per 1000 population and hospital beds per 1000 population in the county of residence. The actual number of months an individual enrolled in Medicaid from March to December 2020 was used as exposure. Stratified multivariate analyses by gender and by age-group (<12 years and 12-<19 years) were also conducted. A priori level of significance was set at $p < 0.05$, and all statistical analyses were performed using STATA version 17/SE. The study was approved by the University of Alabama at Birmingham Institutional Review Board.

## Results

Of 637,792 pediatric Medicaid enrollees during March-December 2020, approximately 16.9% had any telehealth use based on the place of service in claims data (Table 1). The proportion with any telehealth use was approximately 19.6% among NHW children, 17.5% for Native American children, 16.0% for NHB children, 10.5% for Hispanic children, 8.8% for Asian children, and 16.33% for 'Other/Unknown' children. The proportion of telehealth use among enrollees in different RUCA designations was similar, ranging from 16.1%-17.1%. Enrollees in Pov_Q1 through Pov_Q3 zip codes had similar rates of telehealth use (17.5%-17.7%), while among those in the highest poverty quartile zip codes, the rate was 15.6%. Among enrollees residing in zip codes in the highest decile of lack of broadband, 15.5% reported any telehealth use compared to 17.1% of enrollees in other zip codes. The pre-covid or "baseline" telehealth use by enrollees for comparable months in 2018 & 2019 (excluded January and February data for both years), and variations in telehealth utilization by race-ethnicity and the other characteristics, is provided in the S1 Appendix.

Results from the multivariate Poisson mixed-effect model showed that, compared to the reference group of NHW children, NHB children 81% as likely, Hispanic children 55% as likely, and Asian children 46% as likely to have any telehealth use ($p < 0.05$ in all cases). Compared to the reference group of urban enrollees, the large rural area enrollees and isolated area enrollees were significantly less likely to use any telehealth (IRR: 0.90 for both areas; $P < 0.05$,), while enrollees in small rural areas had statistically similar rates (IRR: 1.01, $p > 0.05$). The results regarding the zip-code poverty quartile were not statistically significant; when compared to the reference group of Pov-Q1, those in Pov_Q2 & Pov_Q3 had statistically higher rates of any telehealth use (IRR: 1.03 and 1.04, respectively, $p > 0.05$), and those in Pov_Q4 had statistically

**Table 1. Variations in any telehealth use by race/ethnicity, RUCA, zip-code level poverty & broadband access among Alabama pediatric Medicaid enrollees.**

| | Sample Size | Poisson Mixed Effect Model[a] | |
|---|---|---|---|
| | n (%) | Incidence Rate Ratio | 95% Confidence Interval |
| Full Sample | 647,930 (16.9) | | |
| Race/Ethnicity | | | |
| Non-Hispanic White | 225,130 (19.6) | Reference | |
| Non-Hispanic Black | 240,560 (16.0) | 0.80** | [0.78, 0.81] |
| Hispanic | 41,091 (10.5) | 0.55** | [0.54, 0.57] |
| Native American | 1,779 (17.5) | 0.97 | [0.86, 1.08] |
| Asian | 5,003 (8.8) | 0.46** | [0.42, 0.51] |
| Other | 134,367 (16.3) | 0.84** | [0.82, 0.85] |
| RUCA Category[b] | | | |
| Urban | 484,999 (17.1) | Reference | |
| Large rural | 89,835 (16.1) | 0.90** | [0.83, 0.97] |
| Small rural | 48,209 (16.9) | 0.95 | [0.86, 1.05] |
| Isolated | 24,887 (16.3) | 0.90** | [0.82, 0.98] |
| Zip-code Poverty Quartile | | | |
| Poverty Quartile 1 | 105,621 (17.6) | Reference | |
| Poverty Quartile 2 | 165,187 (17.5) | 1.03 | [0.96, 1.11] |
| Poverty Quartile 3 | 180,246 (17.7) | 1.04 | [0.97, 1.12] |
| Poverty Quartile 4 | 196,747 (15.6) | 0.98 | [0.91, 1.07] |
| No Broadband Connectivity Decile | | | |
| First-ninth decile | 579,446 (17.1) | Reference | |
| Highest decile | 68,484 (15.5) | 0.95 | [0.88, 1.02] |

[a] Poisson mixed-effects model was estimated at the zip-code level with robust error variance. The exposure variable was months enrolled in Medicaid from March to December 2020. Additional control variables included in the model were self-reported gender, age group (0–3 years, 3–6 years, 6–12 years, 12–18 years), pediatricians/family medicine practitioners per 1000 population, and hospital beds per 1000 population in the county of residence.

[b] RUCA: Rural-urban commuting area

[c] Zip-Code level no broadband connectivity decile

**:$P < 0.05$

lower rates of any telehealth use (IRR: 0.98, $p > 0.05$). Those residing in zip-codes in the highest decile of lack of broadband were not significantly associated with telehealth utilization as those living in all other zip-codes (IRR: 0.95, $p > 0.05$). Notably, variance inflation factor analysis did not find evidence of a high degree of multicollinearity between RUCA designation, poverty quartiles, or lack of broadband access. Analyses stratified by gender (Table 2) yielded very similar results for males and females with the exception that those residing in zip-codes in the highest decile of lack of broadband were significantly associated and slightly less likely to use any telehealth as those living in all other zip-codes (IRR:0.91 and IRR:0.93, respectively for females and males; $p < 0.05$). Analyses by age-group (Table 3) also find largely similar results for enrollees under 12 years and 12-<19 years though, once again, enrollees living in the highest decile of lack of broadband were less likely to use any telehealth as those living in all other zip-codes (IRR:0.92 for enrollees under 12 years and 12-<19; $p < 0.05$).

## Discussion

During the COVID-19 pandemic, telehealth became a major vehicle for delivering healthcare services. The COVID-19 PHE declaration, under which CMS made provisions to expand telehealth services, is set to expire soon. However, there is a robust discussion on how telehealth

**Table 2. Variations in any telehealth use by race/ethnicity, RUCA, zip-code level poverty & broadband access among pediatric Medicaid enrollees, stratification by gender.**

| | Poisson Mixed Effects Model (Female)[a] | | Poisson Mixed Effects Model (Male)[a] | |
|---|---|---|---|---|
| | Incidence Rate Ratio | 95% Confidence Interval | Incidence Rate Ratio | 95% Confidence Interval |
| Full Sample | | | | |
| Race/Ethnicity | | | | |
| Non-Hispanic White | Reference | | Reference | |
| Non-Hispanic Black | 0.79** | [0.78, 0.80] | 0.80** | [0.79, 0.82] |
| Hispanic | 0.57** | [0.56, 0.59] | 0.58** | [0.56, 0.60] |
| Native American | 0.93 | [0.83, 1.04] | 0.95 | [0.84, 1.06] |
| Asian | 0.47** | [0.43, 0.52] | 0.46** | [0.42, 0.51] |
| Other | 0.88** | [0.86, 0.89] | 0.89** | [0.87, 0.90] |
| RUCA Category[b] | | | | |
| Urban | Reference | | Reference | |
| Large rural | 0.89** | [0.82, 0.96] | 0.89** | [0.82, 0.96] |
| Small rural | 0.95 | [0.86, 1.05] | 0.94 | [0.85, 1.04] |
| Isolated | 0.89** | [0.82, 0.98] | 0.90** | [0.82, 0.98] |
| Zip-code Poverty Quartile | | | | |
| Poverty Quartile 1 | Reference | | Reference | |
| Poverty Quartile 2 | 1.03 | [0.96, 1.12] | 1.02 | [0.95, 1.10] |
| Poverty Quartile 3 | 1.03 | [0.96, 1.11] | 1.02 | [0.95, 1.10] |
| Poverty Quartile 4 | 0.97 | [0.90, 1.05] | 0.97 | [0.89, 1.05] |
| No Broadband Connectivity Decile | | | | |
| First-ninth decile | Reference | | Reference | |
| Highest decile | 0.91** | [0.85, 0.98] | 0.93** | [0.86, 1.00] |

[a] Poisson mixed-effects model was estimated at the zip-code level with robust error variance. The exposure variable was months enrolled in Medicaid from March to December 2020. Additional control variables included in the model were age group (0–3 years, 3–6 years, and 6–12 years), pediatricians/family medicine practitioners per 1000 population, and hospital beds per 1000 population in the county of residence.

[b] RUCA: Rural-urban commuting area

[c] Zip-Code level no broadband connectivity decile

**:P<0.05

can continue to be a vital part of healthcare delivery and help improve health equity [26]. Thus, it is imperative to understand existing disparities in telehealth use, including among pediatric patients. This is one of the first assessments of telehealth use in a publicly insured pediatric population during the COVID-19 pandemic and explores differences in any telehealth use by race/ethnicity and place of residence. This study focuses on Alabama, a state that struggles with poor performance on several health metrics and saw increase in telehealth utilization for overall population but declines in overall healthcare use and in Emergency Department visits for publicly insured children during the COVID-19 pandemic [14, 27].

Overall, approximately 16.9% of pediatric Alabama Medicaid enrollees used telehealth at least once from March to December 2020 which represents a substantial increase from the 0.16% telehealth utilization rate during comparable periods in 2018 and 2019. The rate of telehealth utilization among Medicaid enrollees during first 10 months of the pandemic is also higher than the 13.4% use of any telehealth in Alabama's stand-alone Children's Health Insurance Program (which covers children 146% to 317% of the Federal Poverty Level) during the same period [15]. This concurs with findings in the literature that low-income populations are more likely to use telehealth than their higher-income counterparts [4, 5]. At the same time,

**Table 3. Variations in any telehealth use by race/ethnicity, RUCA, zip-code level poverty & broadband access among pediatric Medicaid enrollees, stratification by age.**

| | Poisson Mixed Effects Model (Age 12–18 years)[a] | | Poisson Mixed Effects Model (Age less than 12 years)[a] | |
|---|---|---|---|---|
| | Incidence Rate Ratio | 95% Confidence Interval | Incidence Rate Ratio | 95% Confidence Interval |
| Full Sample | | | | |
| Race/Ethnicity | | | | |
| Non-Hispanic White | Reference | | Reference | |
| Non-Hispanic Black | 0.78** | [0.77, 0.80] | 0.81** | [0.79, 0.82] |
| Hispanic | 0.55** | [0.53, 0.57] | 0.59** | [0.57, 0.61] |
| Native American | 0.90 | [0.80, 1.01] | 0.98 | [0.87, 1.09] |
| Asian | 0.46** | [0.42, 0.51] | 0.47** | [0.43, 0.51] |
| Other | 0.83** | [0.82, 0.85] | 0.91** | [0.90, 0.93] |
| RUCA Category[b] | | | | |
| Urban | Reference | | Reference | |
| Large rural | 0.89** | [0.82, 0.96] | 0.89** | [0.82, 0.96] |
| Small rural | 0.94 | [0.85, 1.05] | 0.94 | [0.85, 1.04] |
| Isolated | 0.90** | [0.82, 0.99] | 0.89** | [0.82, 0.98] |
| Zip-code Poverty Quartile | | | | |
| Poverty Quartile 1 | Reference | | Reference | |
| Poverty Quartile 2 | 1.03 | [0.96, 1.11] | 1.02 | [0.95, 1.10] |
| Poverty Quartile 3 | 1.03 | [0.96, 1.11] | 1.02 | [0.95, 1.10] |
| Poverty Quartile 4 | 0.98 | [0.90, 1.06] | 0.96 | [0.89, 1.04] |
| No Broadband Connectivity Decile | | | | |
| First-ninth decile | Reference | | Reference | |
| Highest decile | 0.92** | [0.86, 0.99] | 0.92** | [0.86, 0.99] |

[a] Poisson mixed-effects model was estimated at the zip-code level with robust error variance. The exposure variable was months enrolled in Medicaid from March to December 2020. Additional control variables included in the model were self-reported gender, pediatricians/family medicine practitioners per 1000 population, and hospital beds per 1000 population in the county of residence.

[b] RUCA: Rural-urban commuting area

[c] Zip-Code level no broadband connectivity decile

**:$P < 0.05$

this is lower than the 22–24% of pediatric telehealth use reported in the Household Pulse Survey [28], though it also reported substantial variation across states.

One of the key disparities in telehealth use that emerged from this study's results is by race/ethnicity. While children of all other races/ethnicities had lower telehealth use rates than NHW children, the differences were especially stark for Hispanic and Asian children, who were only about half as likely to use telehealth as NHW children. In contrast, NHB children 80% as likely to use any telehealth compared to NHW children. This strongly suggests that language or communication barriers may have impeded telehealth services. Prior research has indicated that Spanish-speaking patients have greater difficulty with telehealth than English-speaking patients [29], A recent roundtable report from the National Committee for Quality Assurance underlined that using a patient's preferred language was a key element in delivering culturally competent care and that one area that providers had struggled with was how to include translators in telehealth visits [30]. This study's findings reinforce the possibility that language and communication are barriers to telehealth use among publicly insured children and highlight the need to devote resources to overcoming this barrier to reduce race/ethnicity disparities in telehealth utilization.

Interestingly, this study found relatively few differences in rates of any telehealth use by place of residence. Enrollees residing in small rural areas showed no significant differences in telehealth use than enrollees in urban areas, and enrollees in large rural areas and isolated areas were only slightly less likely (90%) to use telehealth than urban residents. It is noted that one California study found that telehealth improved access for publicly insured pediatric patients who lived long distances away from hospitals [31]. Finally, and perhaps surprisingly, lack of broadband access had no significant association with telehealth use.

The research team acknowledges several limitations. Since the analyses are based on claims data, for enrollees with no telehealth use, it cannot be determined whether this was because of a lack of need for health services, language barriers, technology barriers, healthcare provider inability to provide telehealth care, or perceptions about telehealth among the patient or provider. However, one survey of parents and guardians of pediatric patients in Alabama conducted in the fall of 2020 suggests that primary barriers included not being given the option of telehealth by the provider, not clearly understanding what telehealth is, and concerns about the usefulness of telehealth [32]. Also, this study focuses on the role of race and place on pediatric telehealth use, but it does not explore potentially complex interplays between enrollee race and community-level factors, which have been shown to exist for other health services like ED use [33]; this is an area that future research on telehealth use should explore. Further, this study's analysis focused on any versus no telehealth use but did not investigate how extensively patients used telehealth or for what health services telehealth was most frequently used. There was no information on whether telehealth encounters were audio only or audio-visual or whether this impacted patient experience and the quality of care. Also, the data only extended to the end of 2020 and cannot inform on trends in 2021 –though preliminary findings from CMS research indicate that rates of telehealth use among pediatric Medicaid patients nationwide remained relatively steady from June 2020 through June 2021 [34]. Finally, this study is based on one state's Medicaid program. However, the research team believes the findings are particularly relevant for other Deep South states which share many of Alabama's socio-economic and demographic characteristics.

In conclusion, if telehealth continues to be a major vehicle for the delivery of healthcare services for Alabama Medicaid, then particular attention must be paid to race/ethnicity disparities–particularly understanding and addressing the barriers that Hispanic and Asian pediatric patients face in accessing telehealth.

## Supporting information

**S1 Appendix.**
(DOCX)

## Author Contributions

**Conceptualization:** Bisakha Sen, Julie McDougal, Anne Brisendine, Van Nghiem, David Becker.

**Data curation:** Bisakha Sen, Md Jillur Rahim, Pradeep Sharma, Nianlan Yang, Ye Liu, David Becker.

**Formal analysis:** Bisakha Sen, Md Jillur Rahim, Pradeep Sharma, Nianlan Yang, Ye Liu.

**Funding acquisition:** Bisakha Sen.

**Investigation:** Bisakha Sen.

**Methodology:** Bisakha Sen, Md Jillur Rahim, Anne Brisendine, David Becker.

**Supervision:** Bisakha Sen.

**Writing – original draft:** Bisakha Sen, Md Jillur Rahim, Anne Brisendine, David Becker.

**Writing – review & editing:** Bisakha Sen, Md Jillur Rahim, Julie McDougal, Pradeep Sharma, Nianlan Yang, Anne Brisendine, Ye Liu, Van Nghiem, David Becker.

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
