## [Decision Letter · Decision Letter 0]

17 Jan 2023

PONE-D-22-31503Telehealth Use Among Pediatric Alabama Medicaid Enrollees, March-December 2020: Variations by Race/Ethnicity & Place of ResidencePLOS ONE

Dear Dr. Rahim,

Thank you for submitting your manuscript to PLOS ONE. After careful consideration, we feel that it has merit but does not fully meet PLOS ONE’s publication criteria as it currently stands. Therefore, we invite you to submit a revised version of the manuscript that addresses the points raised during the review process.

We look forward to receiving your revised manuscript.

Kind regards,

Kevin Lu, PhD

Academic Editor

PLOS ONE

Journal Requirements:

Reviewers' comments:

Reviewer's Responses to Questions

**Comments to the Author**

1. Is the manuscript technically sound, and do the data support the conclusions?

Reviewer #1: Yes

2. Has the statistical analysis been performed appropriately and rigorously? 

Reviewer #1: Yes

3. Have the authors made all data underlying the findings in their manuscript fully available?

Reviewer #1: Yes

4. Is the manuscript presented in an intelligible fashion and written in standard English?

Reviewer #1: Yes

5. Review Comments to the Author

Reviewer #1: This is a nice paper with new results. It has a large sample and robust methods. Writing is also good.

Here are areas for improvement (Most of them about nuances):

1- Genders differ in all processes related to health care use. In this paper, however, gender differences in correlations are not shown. Only main effects of gender are shown. This is like gender is a confounder or a covariate with only main effects. Some of the effects of gender are indirect through changing pathways.

2- Age groups differ in correlates of health care use in part because differences in their health literacy and chronic disease. Thus, show us if older people and older-old people differ in these paths.

3- Race and ethnic groups differ in correlates of health care use. This means race and ethnicity are not just confounders but also moderators of the SES effects.

Based on minorities' diminished returns, SES has stronger effects on health and health care use of Whites than men. That is, due to racism and social stratification, SES loses some of its effect for marginalized people particularly Black people.

Some examples are here:

https://pubmed.ncbi.nlm.nih.gov/32783022/

https://pubmed.ncbi.nlm.nih.gov/32457934/

https://pubmed.ncbi.nlm.nih.gov/32190811/

4- This paper only focuses on patients data. Neighborhood, health care system, health care provider, and geographic distribution of resources are not even considered.

After these issues are addressed, I will be happy to recommend acceptance.

6. PLOS authors have the option to publish the peer review history of their article (what does this mean?). If published, this will include your full peer review and any attached files.

Reviewer #1: No

---

## [Author Response · Author response to Decision Letter 0]

10 Feb 2023

Here are areas for improvement (Most of them about nuances):

1- Genders differ in all processes related to health care use. In this paper, however, gender differences in correlations are not shown. Only main effects of gender are shown. This is like gender is a confounder or a covariate with only main effects. Some of the effects of gender are indirect through changing pathways.

Thank you for your positive comments on your paper and for your helpful suggestions. We agree that gender may change pathways of associations between race and place with telehealth use; hence, we now also present results that are stratified by gender in table 2.

2- Age groups differ in correlates of health care use in part because differences in their health literacy and chronic disease. Thus, show us if older people and older-old people differ in these paths.

Thank you for your comment. We agree that age may change pathways of associations between race and place with telehealth use. Since we focus on the pediatric population (<19 years), the distinction between older and older-old is not directly relevant, but we believe differences may exist between pre-teen and teen children since the latter may have more of a “say” in their own health utilization decisions. Hence, we stratify results by <12 years and 12-<19 years. 

3- Race and ethnic groups differ in correlates of health care use. This means race and ethnicity are not just confounders but also moderators of the SES effects.

Based on minorities' diminished returns, SES has stronger effects on health and health care use of Whites than men. That is, due to racism and social stratification, SES loses some of its effect for marginalized people particularly Black people.

Some examples are here:

https://pubmed.ncbi.nlm.nih.gov/32783022/

https://pubmed.ncbi.nlm.nih.gov/32457934/

https://pubmed.ncbi.nlm.nih.gov/32190811/

Thank you for your comment. To be clear, we do not treat race and ethnicity as confounders. Rather, they are the main exposure variables of interest, and we are interested in exploring racial and ethnic disparities (as well as disparities by place of residence) in telehealth utilization. That said, future research should explore the interplay of race/ethnicity and place in various telehealth service utilization, and we now mention this in the Limitations and state that this is an area for future research.

Also, thank you for making suggestions regarding additional citations. Since they pertained to the broader adult population while our study focuses on the publicly-insured pediatric population, not all of them were very directly relevant. However, we did cite https://pubmed.ncbi.nlm.nih.gov/32457934/ in the introduction as one example of disparities by race across education levels in the existing literature.

4- This paper only focuses on patient data. Neighborhood, health care system, health care provider, and geographic distribution of resources are not even considered.

We agree that neighborhood, health care system, health care provider, and geographic distribution of resources are important, and indeed we accounted for several of those in our model. Specifically, we included zip-code level poverty quartiles, zip-code level broadband access (especially relevant for telehealth), and family medicine or pediatric physicians per 1000 population in the county. Further, in response to the reviewer’s comment about healthcare systems, we also included hospital beds per 1000 in the models. These additional variables are listed clearly in the notes below in Tables 1, 2, and 3 as model control variables. 

After these issues are addressed, I will be happy to recommend acceptance.

Thank you. We believe we have been able to address the issues that you raised and thank you for your thoughtful comments.

---

## [Decision Letter · Decision Letter 1]

20 Mar 2023

PONE-D-22-31503R1Telehealth Use Among Pediatric Alabama Medicaid Enrollees, March-December 2020: Variations by Race/Ethnicity & Place of ResidencePLOS ONE

Dear Dr. Sen,

Thank you for submitting your manuscript to PLOS ONE. After careful consideration, we feel that it has merit but does not fully meet PLOS ONE’s publication criteria as it currently stands. Therefore, we invite you to submit a revised version of the manuscript that addresses the points raised during the review process.

We look forward to receiving your revised manuscript.

Kind regards,

Kevin Lu, PhD

Academic Editor

PLOS ONE

Reviewers' comments:

Reviewer's Responses to Questions

**Comments to the Author**

1. If the authors have adequately addressed your comments raised in a previous round of review and you feel that this manuscript is now acceptable for publication, you may indicate that here to bypass the “Comments to the Author” section, enter your conflict of interest statement in the “Confidential to Editor” section, and submit your "Accept" recommendation.

Reviewer #2: All comments have been addressed

2. Is the manuscript technically sound, and do the data support the conclusions?

Reviewer #2: Partly

3. Has the statistical analysis been performed appropriately and rigorously? 

Reviewer #2: No

4. Have the authors made all data underlying the findings in their manuscript fully available?

Reviewer #2: No

5. Is the manuscript presented in an intelligible fashion and written in standard English?

Reviewer #2: Yes

6. Review Comments to the Author

Reviewer #2: This is an interesting cross-sectional study aimed to investigate telehealth usage variation by race/ethnicity and place of residence for the pediatric enrollees of the Alabama Medicaid plan. The study found racial disparities as well as the rural-urban disparities in using telehealth among the pediatric enrollees of the Alabama Medicaid plan. The language of the manuscript is also well-written. However, I have some concerns and comments, and I have listed them below:

Major concerns:

1. The measurements of the variables are not clearly described. The authors only talked about what the variables were included, but did not explain how they were categorized or applied.

a. For example, the patients mentioned the RUCA codes was used to categorize the residence areas but did not elaborate how the specific codes were applied. They also did not mention the version of the RUCA codes, and there is no reference for the RUCA code, which could be confusing for the readers who are not familiar with this field.

b. Similarly, what’s the rationale of using quartiles of zip-code level poverty. Is there any reference?

2. There is no data and baseline information on the patients, especially the information on confounders is missing.

3. Is it possible to conduct a Poisson mixed effects model, as there might be fixed differences between residence areas but homogeneity within residence areas?

4. Is it possible to consider the impact of the time, as the usage of telehealth in March and in December could be different?

Minor comments:

1. Is there a specific reason why the study period was only in 2020?

2. What was the telehealth use rate before the pandemic? Is there any published literature about this?

3. Enrollees in Pov_Q3 were more likely to use telehealth, while those in Pov_Q4 were less likely. What’s the potential reason for this difference?

4. Tables: Does “Other” of Race/Ethnicity means both other and unknown?

7. PLOS authors have the option to publish the peer review history of their article (what does this mean?). If published, this will include your full peer review and any attached files.

Reviewer #2: No

---

## [Author Response · Author response to Decision Letter 1]

25 Apr 2023

We have uploaded a document listing our response to the reviewer comments. We are also copying and pasting them here.

We are grateful to the reviewer for the detailed and valuable feedback provided. We are also grateful that the reviewer stated that we adequately addressed all points raised in the prior review.

Below, we outline specific changes we made, with reviewer comments in plain text followed by our response in bold text. 

Major concerns:

1. The measurements of the variables are not clearly described. The authors only talked about what the variables were included, but did not explain how they were categorized or applied.

a. For example, the patients mentioned the RUCA codes was used to categorize the residence areas but did not elaborate how the specific codes were applied. They also did not mention the version of the RUCA codes, and there is no reference for the RUCA code, which could be confusing for the readers who are not familiar with this field.

Thank you for this comment. We have used the latest RUCA codes, revised and released in July 2019, and provided details about RUCA in the methods section, including the citation for the most recent version of RUCA codes (‘Materials and Methods’ subsection, para 2). 

b. Similarly, what’s the rationale of using quartiles of zip-code level poverty. Is there any reference?

We have included citations that support using zip-code level poverty quartiles as a covariate in our model (‘Materials and Methods’ subsection, para 2). Examples include: 

https://www.sciencedirect.com/science/article/pii/S0022480415003510

https://acsjournals.onlinelibrary.wiley.com/doi/10.1002/cncr.21732

https://journals.sagepub.com/doi/10.1177/107327480901600210

2. There is no data and baseline information on the patients, especially the information on confounders is missing.

We agree that this would be useful information for the reader. We have included the baseline information on variation in telehealth utilization by the enrollee characteristics during comparable periods in 2018 & 2019 in the Appendix. Note that this also helps address the reviewer’s later question about telehealth use before the pandemic.

3. Is it possible to conduct a Poisson mixed effects model, as there might be fixed differences between residence areas but homogeneity within residence areas?

In response to this suggestion, we have changed our model to Poisson mixed effects model. 

4. Is it possible to consider the impact of the time, as the usage of telehealth in March and in December could be different?

The number of months an enrollee was in the program is used as an exposure variable in the model. Since our research question is not focused on monthly trends per se, we did not report on monthly rates of telehealth use in this paper, but we have cited a CMS source that reports monthly trends in Medicaid use in telehealth for the country as a whole (https://www.medicaid.gov/state-resource-center/downloads/covid19-data-snapshot-11122021.pdf) 

Minor comments:

1. Is there a specific reason why the study period was only in 2020?

We selected this study period to understand the initial impact of the pandemic on telehealth utilization among pediatric enrollees of the Alabama Medicaid program. There is a substantial lag in time before we receive the completed Alabama Medicaid claims data for any calendar year; hence using data for the latter part of the pandemic was not feasible for this manuscript, though we hope to investigate that in future research.

2. What was the telehealth use rate before the pandemic? Is there any published literature about this?

As mentioned earlier, we have now reported telehealth use among Alabama’s pediatric Medicaid population in 2018 and 2019 in the appendix. To the best of our knowledge, there is no published literature about the telehealth utilization rate of pediatric enrollees in the Alabama Medicaid program. However, we have provided citations that provide information on the telehealth utilization trends among all Medicaid enrollees.

https://www.ncbi.nlm.nih.gov/pmc/articles/PMC9290332/

3. Enrollees in Pov_Q3 were more likely to use telehealth, while those in Pov_Q4 were less likely. What’s the potential reason for this difference?

In the updated mixed effects models, we do not find any statistically significant association between poverty quartiles and telehealth utilization. Thus, we have updated the Results and Discussion sections to reflect this.

4. Tables: Does “Other” of Race/Ethnicity means both other and unknown?

Thank you for this question. We have now clarified that” Other” includes both “other and unknown” and the method section of the manuscript.

---

## [Editor Report · Decision Letter 2]

8 Jun 2023

Telehealth Use Among Pediatric Alabama Medicaid Enrollees, March-December 2020: Variations by Race/Ethnicity & Place of Residence

PONE-D-22-31503R2

Dear Dr. Sen,

We’re pleased to inform you that your manuscript has been judged scientifically suitable for publication and will be formally accepted for publication once it meets all outstanding technical requirements.

Kind regards,

Kevin Lu, PhD

Academic Editor

PLOS ONE

---

## [Editor Report · Acceptance letter]

15 Jun 2023

PONE-D-22-31503R2 

Telehealth Use Among Pediatric Alabama Medicaid Enrollees, March-December 2020: Variations by Race/Ethnicity & Place of Residence 

Dear Dr. Sen:

I'm pleased to inform you that your manuscript has been deemed suitable for publication in PLOS ONE. Congratulations! Your manuscript is now with our production department. 

Kind regards, 

on behalf of

Professor Kevin Lu 

Academic Editor

PLOS ONE